# KIF13B-mediated VEGFR2 trafficking is essential for vascular leakage and metastasis in vivo

Stephen B Waters[1], Joseph R Dominguez[1], Hyun-Dong Cho[1], Nicolene A Sarich[1], Asrar B Malik[1], Kaori H Yamada[1,2]

**VEGF-A induces vascular leakage and angiogenesis via activating the cell surface localized receptor VEGF receptor 2 (VEGFR2). The amount of available VEGFR2 at the cell surface is however tightly regulated by trafficking of VEGFR2 by kinesin family 13 B (KIF13B), a plus-end kinesin motor, to the plasma membrane of endothelial cells (ECs). Competitive inhibition of interaction between VEGFR2 and KIF13B by a peptide kinesin-derived angiogenesis inhibitor (KAI) prevented pathological angiogenesis in models of cancer and eye disease associated with defective angiogenesis. Here, we show the protective effects of KAI in VEGF-A-induced vascular leakage and cancer metastasis. Using an EC-specific KIF13B knockout (*Kif13b^iECKO*) mouse model, we demonstrated the function of EC expressed KIF13B in mediating VEGF-A-induced vascular leakage, angiogenesis, tumor growth, and cancer metastasis. Thus, KIF13B-mediated trafficking of VEGFR2 to the endothelial surface has an essential role in pathological angiogenesis induced by VEGF-A, and is therefore a potential therapeutic target.**

## Introduction

Metastasis is the most common cause of cancer-related mortality (Pari et al, 2020). Metastatic cancer cells spread by crossing the endothelial barriers of blood and lymphatic vessels, thus modulating the barrier function of endothelial cells (ECs) is likely important in preventing metastasis. Blood and lymphatic microvessels are composed of a monolayer of ECs, as well as mural pericyte localized in the abluminali side of the endothelium. ECs form a barrier, a gatekeeper for transmigrating cells (Duong & Vestweber, 2020; Eelen et al, 2020). VEGF-A is crucial for disrupting the endothelial barrier function and also mediates angiogenesis (Apte et al, 2019). Excessive VEGF-A generation, however, causes unfettered vascular leakage and angiogenesis by binding and activating receptor tyrosine kinase VEGFR2 localized on the surface of ECs (Adams & Alitalo, 2007; Simons et al, 2016; Apte et al, 2019). Besides the role of VEGF-A in pathological angiogenesis, it also induces physiological angiogenesis essential for

normal vertebrate development (Apte et al, 2019; Eelen et al, 2020). *Vegf-A* heterozygous (+/−) (Carmeliet et al, 1996; Ferrara et al, 1996), *Vegfr2* null (Shalaby et al, 1995), and *Y1173F-Vegfr2* (Sakurai et al, 2005) mice are embryonically lethal because of severely impaired vascular development. In contrast, *Vegfr2^+/−* mice are viable with normal vessel formation (Oladipupo et al, 2018). Surprisingly, they also show less tumor angiogenesis and tumorigenesis (Oladipupo et al, 2018) indicating the importance of the fine-tuning of VEGFR2 signaling in driving either physiological or pathological angiogenesis (Simons et al, 2016; Apte et al, 2019). Thus, decreasing the signaling function of VEGFR2 by half reduced tumorigenesis without affecting the essential function of VEGF signaling in the maintenance of healthy blood vessels.

The availability of VEGFR2 on the cell surface at any given time is a critical factor regulating VEGFR2 signaling. On the EC surface, VEGFR2 is normally in a dephosphorylated inactive state in the quiescent endothelium (Simons et al, 2016). Upon binding to VEGF-A, VEGFR2 is phosphorylated at $Tyr^{1175}$ ($Tyr^{1173}$ in mice), resulting in receptor internalization (Simons et al, 2016), trafficking to endosomes, and activation of downstream Src and Erk signaling (Lanahan et al, 2010; Simons et al, 2016). VEGFR2 is then sorted for either degradation or recycling to the cell surface (Simons et al, 2016) to make the receptor available for another round of ligand binding (Lanahan et al, 2010; Manickam et al, 2011; Tiwari et al, 2013). We previously showed that the kinesin 3 family molecular motor KIF13B was required for transporting VEGFR2 to the cell surface (Yamada et al, 2014). Interestingly, tumor angiogenesis and tumor growth were both inhibited by interfering with the interaction between KIF13B and VEGFR2 by the peptide kinesin-derived angiogenesis inhibitor (KAI) (Yamada et al, 2017). Inhibiting VEGFR2 trafficking by KAI suppressed localization of VEGFR2 on the cell surface, and reduced VEGF/VEGFR2 signaling. As VEGF/VEGFR2 signaling is also critical for maintaining healthy ECs, there is a concern for the systemic effect of anti-VEGF therapy (Crawford & Ferrara, 2009; Ferrara & Adamis, 2016). Unlike the anti-VEGF strategy, the inhibition of VEGFR2 trafficking does not affect the overall expression of VEGFR2 nor the function of VEGFR2 to maintain the healthy ECs (Yamada et al, 2017). Thus, selectively targeting VEGFR2 trafficking may be a safe alternative to prevent VEGF/VEGFR2-mediated

[1]Department of Pharmacology and Regenerative Medicine, University of Illinois College of Medicine, Chicago, IL, USA  [2]Department of Ophthalmology and Visual Sciences, University of Illinois College of Medicine, Chicago, IL, USA

Correspondence: horiguch@uic.edu

angiogenesis and angiogenesis-mediated diseases such as cancer. Global KIF13B knockout (*Kif13b^KO*) mice were viable and showed no apparent developmental changes; the only defect was increased circulating cholesterol levels (Kanai et al, 2014). As *Kif13b^KO* showed reduced pathological angiogenesis in the mouse model of wet age-related macular degeneration (Waters et al, 2021), the effects of KIF13B deficiency can be limited to preventing pathological angiogenesis, similar to *Vegfr2^+/−* mice (Oladipupo et al, 2018). To address the role of KIF13B on VEGFR2 trafficking in mediating vascular leakage, and angiogenesis in vivo, we generated an EC-specific inducible knockout of KIF13B (*Kif13b^iECKO*). We demonstrated the fundamental role of KIF13B-mediated VEGFR2 trafficking in the mechanism of cancer metastasis. Our results provide proof-of-concept that targeting VEGFR2 trafficking to the plasma membrane may be a valid anti-metastatic strategy.

# Results

## Inhibition of VEGFR2 trafficking prevents cancer metastasis

We previously showed that our peptide KAI, an inhibitor for VEGFR2 trafficking, inhibits tumor angiogenesis and tumor growth (Yamada et al, 2017). To test whether KAI has any effect on cancer metastasis, we used a lung metastasis model using i.v. injection of B16F10 melanoma via tail vein (Fig 1A–D). After injection of B16F10 in C57BL mice via tail vein, mice received injection of control peptide (ctrl) or KAI (10 mg/kg b.w.) three times per week for 2 wk. Lungs were isolated and metastatic foci were counted (Fig 1A and B). The lungs were further analyzed with H&E staining and immunostaining with Ki67 antibody (Fig 1C and D). The lung metastasis of B16F10 melanoma was significantly reduced by treatment with KAI.

Cancer cells secrete VEGF, which can induce vascular leakage (Reymond et al, 2013). Thus, we wonder whether this inhibitory effect of KAI is through the inhibition of VEGFA-induced vascular leakage. Then the efficacy of KAI in VEGF-A-induced vascular leakage was tested by the Miles Evans blue-albumin extravasation assay (Fig 1E and F). After i.v. injection of Evans blue, C57BL6 mice received s.c. injection of VEGF-A mixed with either ctrl or KAI in the back. Compared with the ctrl, KAI significantly inhibited VEGF-A-induced vascular leakage (Fig 1E and F). Taken together, these data suggest the trafficking of VEGFR2 mediated by KIF13B is important to regulate vascular leakage and metastasis.

## Generation of EC-specific inducible *Kif13b* knockout mice

To further confirm the above finding, we used the genetic approach of deleting KIF13B. As global *Kif13b* knockout (*Kif13b^KO*) showed a defective phenotype in liver hepatocytes (Kanai et al, 2014) and pathological angiogenesis in choroidal vasculature (Waters et al, 2021), we tested the expression levels of KIF13B in ECs and liver hepatocytes by Western blotting and real-time PCR (Fig 2A and B). We observed that KIF13B was expressed in primary culture of mouse lung ECs and human umbilical vein endothelial cells (HUVEC) as well as in liver hepatocyte HepG2 (Fig 2A), whereas its expression was relatively low in human smooth muscle cells (Fig 2B). Interestingly,

the expression of KIF13B was up-regulated by VEGF-A stimulation in cultured mouse ECs and HUVEC (Fig 2A), suggesting that VEGF-A can itself control KIF13B expression.

To address the role of KIF13B in ECs in vivo, we generated mice in which KIF13B was conditionally deleted in the endothelium. *Kif13b*-floxed mice from the European Mouse Mutant Archive were crossed with flippase1 (FLP1)-expressing mice (The Jackson Laboratory), followed by crossing with *endo-SCL-Cre-ER^T* mice (Gothert et al, 2004) (Fig S1A). We confirmed KIF13B was deleted in ECs isolated from tamoxifen-induced *Kif13b^tm1c/tm1c, endo-SCL-Cre (+)*, whereas KIF13B was normally expressed in ECs from *Kif13b^tm1c/tm1c* (Fig 2C) (referred to as *Kif13b^iECKO* and *Kif13b^WT* hereafter).

## Defective VEGF-A responses in endothelial-specific inducible *Kif13b* knockout mice

To study the role of KIF13B in the VEGF-induced function of EC, we carried out the Matrigel plug assay with embedded VEGF-A in *Kif13b^iECKO* and *Kif13b^WT* mice (Fig 2D). *Kif13b^WT* developed capillary vessels, whereas their formation was significantly reduced in *Kif13b^iECKO* (Fig 2D and E). The capillary formation was also not observed without VEGF-A in both *Kif13b^iECKO* and *Kif13b^WT* (Fig 2D and E).

We also used the ex vivo aortic ring assay in which the rings were isolated from *Kif13b^iECKO* and *Kif13b^WT* littermates and incubated in collagen gel supplemented with VEGF-A. We observed VEGF-A-induced EC sprouting in *Kif13b^WT*, whereas the response was defective in *Kif13b^iECKO* (Fig 2F and G). These data suggest the important role of KIF13B in the VEGF-A-induced response of EC in vivo and ex vivo.

## In vivo role of KIF13B in tumor angiogenesis

We also determined the role of KIF13B in cancer growth and angiogenesis by stably transfecting B16F10 melanoma with firefly luciferase (*luc2*) and injecting these cells s.c. in the right flank of *Kif13b^iECKO* and *Kif13b^WT*. Tumor growth was monitored by bioluminescence using in vivo imaging system (IVIS), and tumor size was measured by calipers (Fig 3A–C). Tumor growth was significantly reduced in *Kif13b^iECKO* as compared with *Kif13b^WT* controls. The effects of KIF13B expression in EC in tumor angiogenesis were examined by immunohistochemistry (IHC) of CD31 in tumor tissue (Fig 3D and E). We observed a large number of capillaries with CD31 staining in tumors of *Kif13b^WT*, whereas the number was significantly decreased in *Kif13b^iECKO* (Fig 3E).

## Endothelial specific deletion of KIF13B in mice prevents VEGF-A–induced vascular leakage and tumor metastasis

We next determined whether deletion of KIF13B contributed to vascular leakage induced by VEGF-A using the Miles Evans blue-albumin extravasation assay (Fig 4A and B). After i.v. injection of Evans blue, *Kif13b^iECKO* and *Kif13b^WT* mice received s.c. injection of PBS and VEGF-A in the back. Vascular leakage was observed post VEGF-A injection in *Kif13b^WT*, whereas the response was significantly impaired in *Kif13b^iECKO*; importantly, the basal leakage value was not altered (Fig 4B). We next determined metastasis of melanoma in the

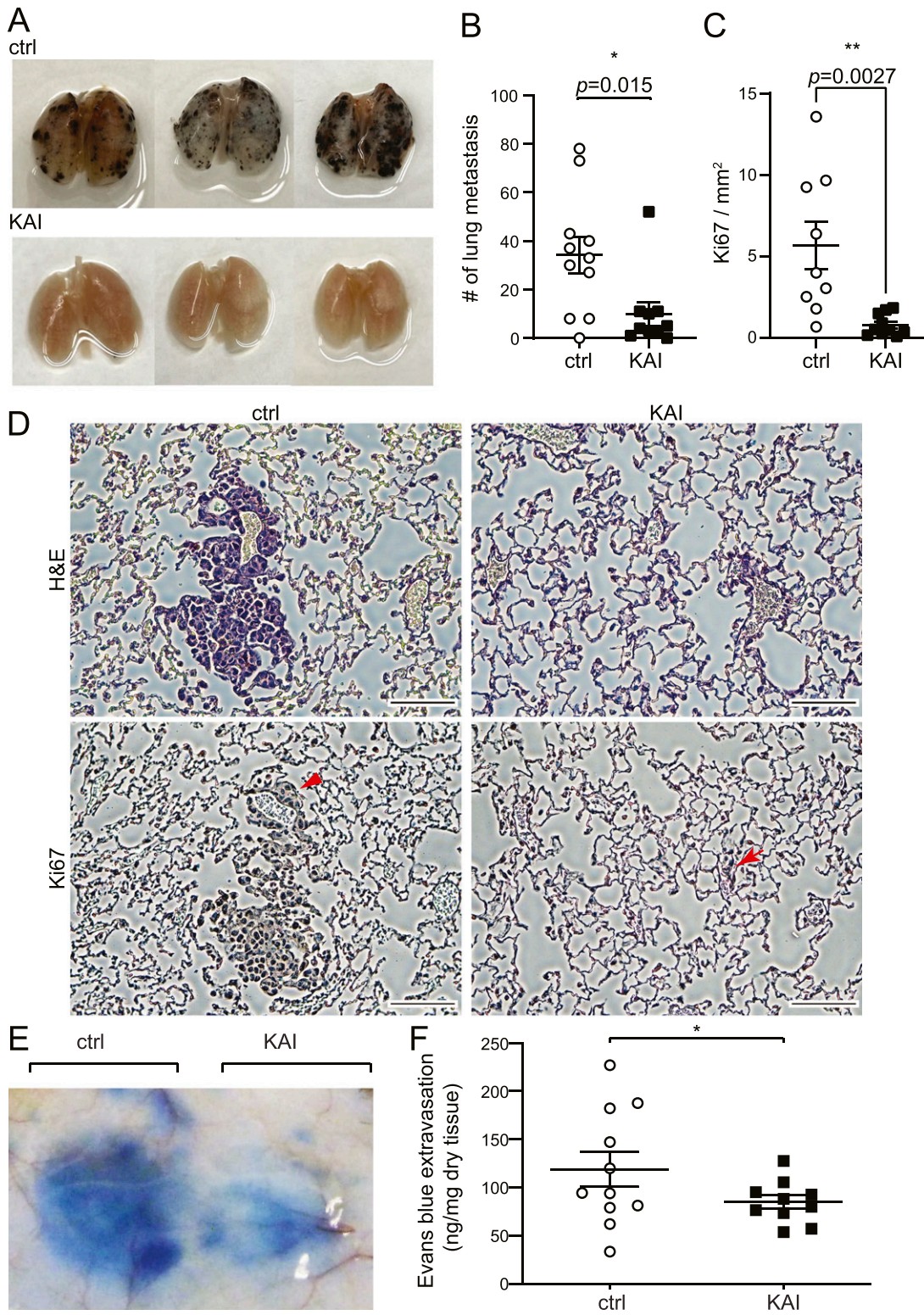

**Figure 1. Inhibition of VEGFR2 trafficking prevents vascular leakage and metastasis in mice.**
**(A, B)** Lung metastasis was examined by pigmented foci of B16F10 melanoma. C57BL/6 received tail vein injection with $1 \times 10^5$ of metastatic B16F10 melanoma cells. Mice were divided into two groups; each group received injections of control peptide or KAI (10 mg/kg b.w.) three times/week for 2 wk, and the lungs were isolated and fixed. The pigmented foci on the lung surface were examined. The plots show the number of metastatic lung foci on day 14. Treatment of KAI significantly decreased lung metastasis compared with the control peptide. N = 11, 10 for control peptide and KAI, respectively. **(C, D)** Representative images of H&E and immunostaining of Ki67 of lungs from mice treated with either control peptide or KAI for 2 wk after injection with $1 \times 10^5$ B16F10 melanoma via tail vein. Scale bar, 100 $\mu$m. Number of Ki67-positive

lungs of *Kif13b^iECKO* and *Kif13b^WT* mice. B16F10 melanoma cells were injected via the tail vein to assess lung metastasis in *Kif13b^WT* and *Kif13b^iECKO*. 2 wk after injection, lungs were isolated, and metastatic foci were counted (Fig 4C–E). We found that *Kif13b^iECKO* had a significantly lower rate of metastatic pulmonary colonization than *Kif13b^WT* controls. Hematoxylin and eosin (H&E) staining and immunostaining with Ki67 antibody further revealed that WT lungs had metastasized foci of melanoma in each lobe, whereas lungs of *Kif13b^iECKO* showed significantly reduced number and size of metastasis (Fig 4D).

To further analyze the specificity of KAI to EC and other cell types, we tested *Kif13b^iECKO* and *Kif13b^WT* controls for treatment with KAI. *Kif13b^iECKO* and *Kif13b^WT* received i.v. injection of B16F10 melanoma, followed by i.p. injection of ctrl peptide or KAI (10 mg/kg b.w.) three times/week for 2 wk. Lungs were isolated, and metastatic foci were counted (Fig 5). Consistent with Fig 1, KAI treatment significantly reduced the number of metastasis in *Kif13b^WT* compared with treatment with ctrl peptide (Fig 5A and B). Consistent with Fig 4, *Kif13b^iECKO* and *Kif13b^WT* with ctrl peptide treatment showed a significant difference. However, KAI treatment in *Kif13b^iECKO* did not show further inhibition, compared with ctrl treatment in *Kif13b^iECKO* or KAI treatment in *Kif13b^WT* (Fig 5A and B). This data suggest that the target of KAI is mainly KIF13B and VEGFR2 in EC.

Taken together, these results demonstrated that KIF13B has an essential role in regulating VEGF-A-induced vascular leakage and angiogenesis by regulating VEGFR2 trafficking. Therefore, inhibition of the VEGFR2 trafficking by KAI can be a promising strategy to prevent cancer metastasis.

## Discussion

VEGF-A-induced angiogenesis requires the trafficking of its cognate receptor VEGFR2 to the EC plasma membrane, where it is ligated by VEGF-A (Lanahan et al, 2010; Nakayama et al, 2013). We previously showed a key role for the plus-end kinesin molecular motor KIF13B as a polarized transporter of VEGFR2 to the plasmalemmal membrane in cultured ECs (Yamada et al, 2014). As a spatial organization of VEGFR2 at the plasma membrane and activation of VEGFR2 pathways may be modulated by blood flow and thus influence downstream signaling pathways (Simons et al, 2016), this study was made to address the in vivo role of KIF13B in mediating angiogenesis and metastasis. Previous studies in global *Kif13b^KO* mice showed that the mice developed normally without any apparent developmental defects (Kanai et al, 2014). However, pathological angiogenesis in blinding eye disease wet age-related macular degeneration was significantly reduced in *Kif13b^KO* (Waters et al, 2021). Thus, the function of KIF13B may be more important in pathological angiogenesis than developmental angiogenesis. To further analyze the function of KIF13B in EC, we generated EC-specific inducible *Kif13b* knockout mice. Our results demonstrated that deletion of KIF13B in ECs severely impaired VEGF-A-induced neovascularization and tumor

angiogenesis. The results demonstrated that KIF13B in EC has a fundamental role in mediating pathological angiogenesis.

As VEGF-A/VEGFR2 signaling is vital for vascular development, deletion of either the ligand or receptor is embryonic lethal (Shalaby et al, 1995). Heterozygous *Vegfr2^+/LacZ*, however, showed normal vascularization of the postnatal retina (Zarkada et al, 2015), indicating that even a small amount of VEGFR2 can support physiological angiogenesis (Zarkada et al, 2015). In contrast, pathological angiogenesis such as tumor angiogenesis was reduced in *Vegfr2^+/−* mice (Oladipupo et al, 2018). Therefore, reducing the amount of functional VEGFR2 can be a promising strategy to inhibit pathological angiogenesis in disease conditions while keeping the small amount of VEGFR2 for maintaining healthy vessels. Our strategy is to reduce the amount of VEGFR2 on the cell surface by inhibiting VEGFR2 trafficking mediated by KIF13B. Knockdown of KIF13B in EC did not affect the total amount of VEGFR2, whereas trafficking of VEGFR2 to the plasma membrane was inhibited (Yamada et al, 2014). Similarly, restoration of cell–surface VEGFR2 after its internalization was reduced by KAI treatment in cultured EC (Yamada et al, 2017). In the present study, EC-specific deletion of KIF13B did not affect the total amount of VEGFR2 (data not shown). Based on the in vitro study, inhibition of KIF13B by genetic depletion or KAI treatment only prevents trafficking of VEGFR2 to the plasmalemma and thereby blocked angiogenesis, suggesting that the primary role of KIF13B is in mediating the trafficking and spatial organization of VEGFR2 at the plasma membrane.

The role of KIF13B as a molecular motor has been demonstrated in several cell types (Hanada et al, 2000; Horiguchi et al, 2006; Yamada et al, 2007, 2014; Hirokawa et al, 2009; Jenkins et al, 2012; Bentley et al, 2015; Noseda et al, 2016; Liao et al, 2018; Yang et al, 2019). KIF13B was shown to regulate cell signaling by transporting proteins involved in cell polarity (Hanada et al, 2000; Horiguchi et al, 2006; Hirokawa et al, 2009; Bentley et al, 2015). In neurons, KIF13B played an important role in axon formation (Horiguchi et al, 2006; Yoshimura et al, 2010) and transport of dendrite-selective vesicles (Jenkins et al, 2012). In glia, KIF13B regulated myelination (Noseda et al, 2016). In ECs, as we showed, KIF13B-mediated directional trafficking of VEGFR2 is responsible for the front-rear polarity necessary for sprouting of ECs. Importantly, quiescent ECs were not affected by KIF13B deletion in EC, suggesting that the proper spatial positioning of VEGFR2 at the plasmalemma as mediated by KIF13B is an essential requirement for angiogenesis.

This study showed that deletion of KIF13B in EC inhibited both VEGF-A-induced angiogenesis and vascular permeability. Pharmacological inhibition of VEGFR2 trafficking by KAI also inhibited VEGF-A-induced angiogenesis (Yamada et al, 2017; Waters et al, 2021) and vascular permeability (this study). Then, how is angiogenesis and vascular leakage correlated? Endothelial adherens junctions (AJs) are essential for maintaining endothelial barrier function and responsible for the contact inhibition property of quiescent confluent endothelial monolayers that restricts

foci/mm² was measured and shown in graph C as mean ± SEM. **(E, F)** Evans blue extravasation showing the effect of KAI on VEGF-A-induced endothelial cell permeability in C57BL/6 mice. C57BL/6 received i.v. injection of Evans blue followed by s.c. injection of VEGF-A and KAI or control peptide to test VEGF-A-induced permeability. Extravasation of Evans blue to skin was measured and plotted in a graph. N = 11 and 10 for control peptide and KAI, respectively.

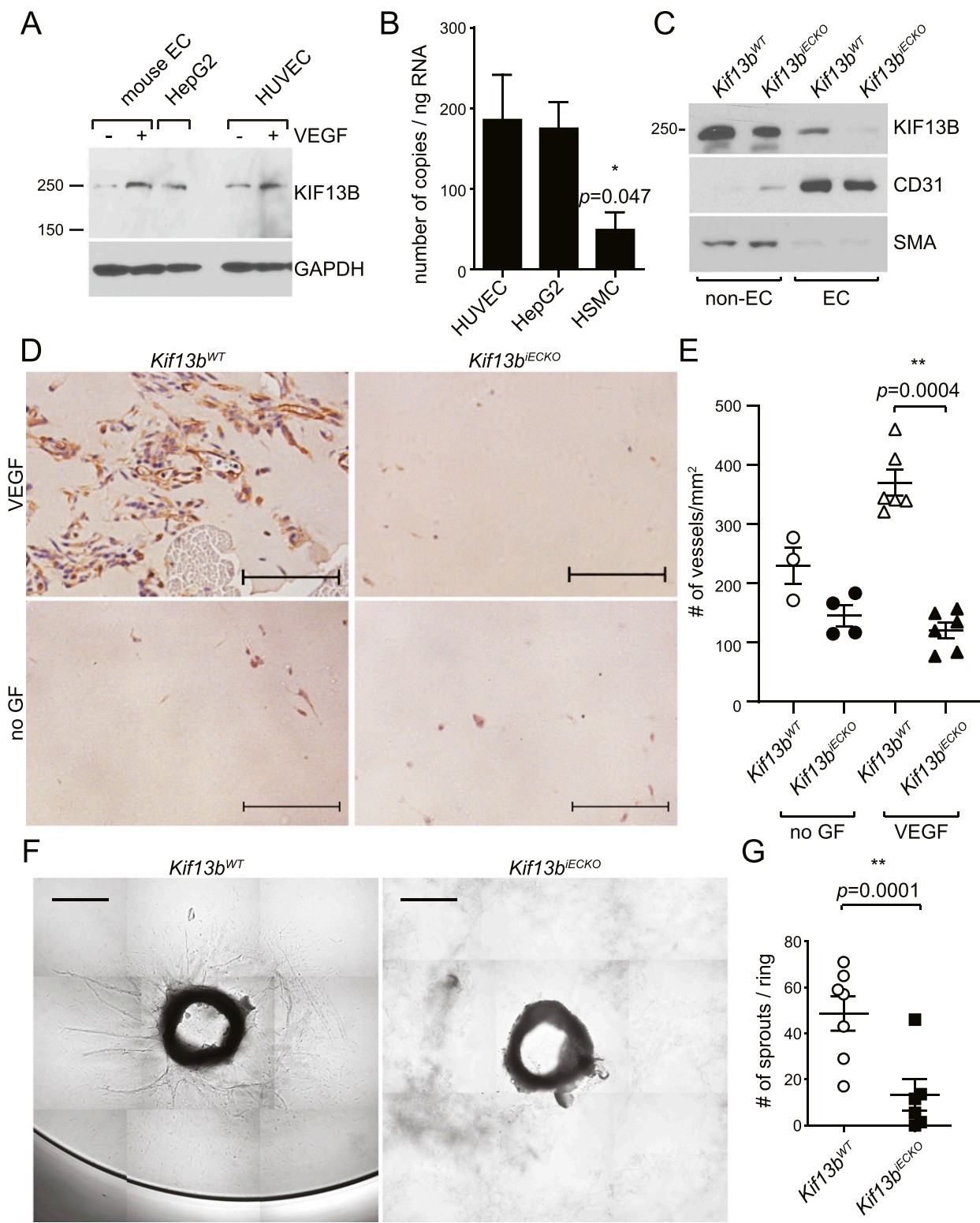

**Figure 2. Endothelial cell-specific knockout of KIF13B in mice prevents VEGF-A–induced neovascularization and sprouting angiogenesis.**
**(A)** Western blots showing expression of endogenous KIF13B in mouse pulmonary endothelial cells (mouse EC) and HUVEC. Hepatocyte HepG2 was used as a control.
**(B)** Real-time PCR revealed that KIF13B expresses 186 ± 56, 175 ± 28, 49 ± 22 copies/ng RNA in HUVEC, HepG2, and human smooth muscle cells, respectively. N = 9, 9, 3, respectively. **(C)** Western blotting showing deletion of KIF13B in ECs from tamoxifen-treated *Kif13b^{tm1c/tm1c endo-SCL-Cre (+)}* mice, whereas ECs from *Kif13b^{tm1c/tm1c}* control mice express KIF13B. ECs were isolated from mice lungs using anti-CD31 antibody and anti-ICAM2 antibody (Jin et al, 2012). The expression of KIF13B (250 kD band, arrowhead) was detected by anti-KIF13B Ab. CD31⁻ cells also show a truncated KIF13B band. A-smooth muscle actin is a marker of smooth muscle cells and myofibroblast. Isolated ECs

angiogenesis (Lampugnani et al, 2006; Giannotta et al, 2013). Thus, disassembly of AJs is required to initiate angiogenesis (Carmeliet & Jain, 2011). In the quiescent confluent endothelium, VEGFR2 is kept inactive through homotypic interaction of VE-cadherin at AJs (Lampugnani et al, 2006; Hayashi et al, 2013). VEGF-A increases endothelial permeability through dissociation of VEGFR2 from VE-cadherin (Weis & Cheresh, 2005; Gavard & Gutkind, 2006) and internalization of both VE-cadherin and VEGFR2 from the plasma membrane (Lampugnani et al, 2006). Internalized VEGFR2 is then either recycled back to the cell surface (Gampel et al, 2006; Manickam et al, 2011; Yamada et al, 2014) or degraded in lysosomes (Simons et al, 2016). Studies showed that VEGFR2 localized at the filopodia of tip cells during angiogenesis to sense the VEGF-A gradient and induce directional migration (Gerhardt et al, 2003; Jakobsson et al, 2010; Hayashi et al, 2013). The trafficking of VEGFR2 in this context is thus essential for angiogenesis (Gampel et al, 2006; Manickam et al, 2011; Nakayama et al, 2013; Yamada et al, 2014, 2017). However, it is not known how VEGFR2 trafficking can affect the transition from quiescence to angiogenic ECs. Because KIF13B, a kinesin motor trafficking along microtubules, transports VEGFR2 to the cell surface (Yamada et al, 2014), directional VEGFR2 trafficking induced by KIF13B is likely important in the accumulation of VEGFR2 at filopodia while decreasing VEGFR2 amount at AJs. Thus, polarized VEGFR2 distribution mediated by KIF13B may be essential for the ECs to transition from quiescent to the angiogenic phenotype. Our finding that inhibition of KIF13B-mediated VEGFR2 trafficking impaired angiogenesis in vivo is consistent with the finding that migrating ECs exhibited fast growth of microtubules at the front of cells and depolymerization at the trailing edge (Hu et al, 2002; Tzima et al, 2003; Braun et al, 2014). Such microtubule dynamics may be a factor in supporting VEGFR2 trafficking to the membrane to establish the front-rear polarity of migrating ECs (Mayor & Etienne-Manneville, 2016). VEGFR2 trafficking likely regulates transition from quiescent EC, short-term barrier relaxation, long-term angiogenesis, and vascular maturation to reestablish the endothelial barrier. However, further investigation is needed to clarify how VEGFR2 trafficking affects signaling pathways to proceed with this transition.

In diseased conditions, such as cancer and blinding eye diseases, newly formed vasculature is kept leaky because of impaired endothelial barrier function caused by excessive VEGF-A (Claesson-Welsh et al, 2020). Anti-VEGF therapy (Ferrara & Adamis, 2016) or inhibitor of VEGFR2 trafficking (Yamada et al, 2017; Waters et al, 2021) is thus effective for cancer and blinding eye diseases. As cancer cells need to extravasate/intravasate the endothelial barrier to metastasize, inhibition of endothelial permeability can be a promising strategy to inhibit metastasis. This study demonstrated that inhibition of KIF13B-mediated VEGFR2 trafficking inhibited VEGF-A-induced pathological angiogenesis, vascular leakage, and thereby tumor metastasis. Although KAI treatment or knockout of *Kif13b*

inhibited metastasis, KAI did not show additional inhibitory effect in *Kif13b^{iECKO}* (Fig 5), suggesting the target of KAI is mainly the function of KIF13B in EC. Metastasis is multiple-step events, such as escape of cancer cells from the primary tumor, intravasation of cancer cells into the bloodstream, survival of cancer cells in the bloodstream, extravasation of cancer cells through the vascular barrier at the distal site, and colonization of cancer cells at the distal site. This experimental setting skips initial steps and starts from the injection of the cancer cells into the bloodstream. And the inhibitory effect of KAI was observed only in the presence of KIF13B in EC (Fig 5), suggesting the main target of KAI is KIF13B-mediated VEGFR2 trafficking in EC, not survival of cancer cells in the bloodstream nor colonizing of cancer cells in the lungs. Further investigation of safety and any adverse effect of KAI is needed for preclinical drug development. Nonetheless, this study showed that targeting KIF13B-mediated VEGFR2 trafficking to the EC plasmalemma may be a potential anti-angiogenesis and anti-metastatic strategy.

# Materials and Methods

## Antibodies

Antibodies used for this study include rabbit antibodies against KIF13B (HPA025-023; Sigma-Aldrich) and GAPDH (G9545; Sigma-Aldrich), CD31 (ab28364; Abcam), and Ki67 (Abcam). Mouse antibody against smooth muscle actin was from Sigma-Aldrich. The rat antibodies used in this study are CD31 (CBL1337; EMD Millipore) and ICAM2 (MAB3267; Abnova). HRP-conjugated secondary antibodies were from Jackson Immunoresearch (715-035-150 and 711-035-152).

## Cells

Mouse melanoma B16F10 (ATCC) were transfected with firefly luciferase (Promega), and stably transfected cells were selected by using G418 (GoldBio). HUVECs were purchased from Lonza and cultured as the manufacture's procedure.

## Knockout mouse model

To obtain EC-specific inducible *Kif13b* knockout mice, *Kif13b*-floxed mice (*Kif13b^{tm1a}* in Fig S1A) from the European Mouse Mutant Archive was crossed with flippase1 (flp1)-expressing mice (The Jackson Laboratory), followed by crossing with *endo-SCL-Cre^{ERT}* mice (Gothert et al, 2004) (Fig S1A). *Kif13b^{tm1a}*, *Kif13b^{WT/tm1c}*, *Kif13b^{tm1c/tm1c}*, and *Kif13b^{tm1c/tm1c, endo-SCL-Cre (+)}* mice were confirmed by PCR (Fig S1B). To confirm EC-specific deletion of KIF13B, we isolated lung EC from tamoxifen-induced *Kif13b^{tm1c/tm1c, endo-SCL-Cre (+)}* and *Kif13b^{tm1c/tm1c}* as CD31^{+}/ICAM2^{+} population using magnetic beads, whereas CD31^{-} cells were non-EC. The purity was confirmed by Western blotting with CD31

---

(CD31^{+}/ICAM2^{+} population) from both mice were CD31^{+} and smooth muscle actin. **(D, E)** VEGF-A–induced neovascularization in Matrigel plug in *Kif13b^{iECKO}* and *Kif13b^{WT}* mice. After tamoxifen treatment, Matrigel supplemented with 40 ng/ml (1.7 nM) VEGF-A and 2 U heparin, or 2 U heparin alone (-growth factor) were injected s.c. in *Kif13b^{iECKO}* and *Kif13b^{WT}* counterpart mice. 7 d after implantation, the Matrigel plugs were collected and stained with anti-CD31 Ab. **(A)** Representative images were shown in (A). Scale bar: 100 μm. **(B)** The number of CD31-positive vessels was counted and shown as mean ± SE in graph (B). N = 3, 4, 6, and 6 for *Kif13b^{WT}* -growth factor (GF), *Kif13b^{iECKO}* -GF, *Kif13b^{WT}* + VEGF-A, and *Kif13b^{iECKO}* + VEGF-A, respectively. **(F, G)** VEGF-A–induced EC sprouting from aortic rings of *Kif13b^{iECKO}* and *Kif13b^{WT}*. The aortic rings were isolated from tamoxifen-treated *Kif13b^{iECKO}* and *Kif13b^{WT}* and embedded in the collagen gels supplemented with VEGF-A. **(C)** VEGF-A–induced sprouting was analyzed as shown in (C). Scale bar: 200 μm. **(G)** The number of sprouting was shown as mean ± SE in (G). N = 7 and 6 for *Kif13b^{WT}* and *Kif13b^{iECKO}*, respectively.

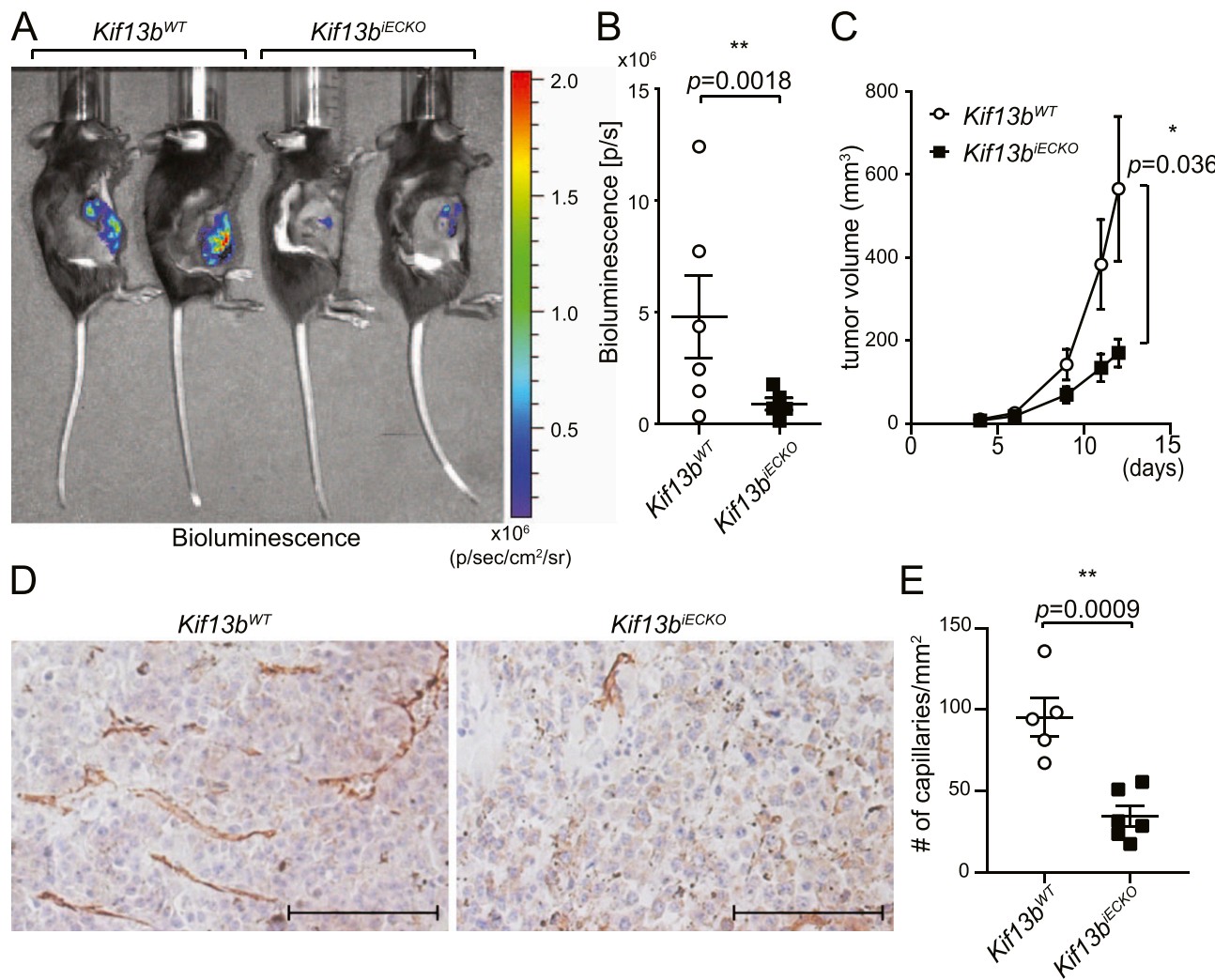

**Figure 3.  Endothelial cell-specific deletion of KIF13B inhibits tumor angiogenesis and tumor growth.**
**(A, B, C)** *Kif13b^iECKO* and *Kif13B^WT* mice were treated with tamoxifen i.p. an injection every second day for 2 wk. 3 wk after the completion of tamoxifen injections, luciferase-expressing B16F10 were injected s.c. at the right flank of these mice. **(A, B)** Tumor growth was monitored by bioluminescence using the IVIS imaging system, the representative images of day 12 were shown in (A), and the graph with mean ± SE was shown in (B). N = 6. *t* test, P = 0.0018. In (C), tumor growth was also monitored by measuring with a caliper, and shown in the graph by mean ± SE. N = 6. **(D, E)** The tumor was isolated at day 12 and stained with CD31 to visualize vascularization in the tumor. **(D)** The representative images were shown in (D). Scale bar: 100 $\mu$m. **(E)** The number of CD31 positive capillaries were counted and shown as mean ± SE in graph (E).

and SMA, respectively (Fig 1C). We confirmed KIF13B was deleted in ECs isolated from tamoxifen-induced *Kif13b^{tm1c/tm1c, endo-SCL-Cre (+)}*, whereas KIF13B was expressed in ECs from *Kif13b^{tm1c/tm1c}* (Fig 2C) and referred to as *Kif13b^iECKO* and *Kif13b^WT* in this study. All animal experiments were carried out to comply with the relevant laws and institutional guidelines and were approved by the Animal Care Committee administered through the Office of Animal Care and Institutional Biosafety at UIC.

## Matrigel plug assay

Growth factor-induced neovascularization was tested in Matrigel plug assay as described previously (Yamada et al, 2017). Briefly, growth factor reduced Matrigel (BD) was mixed with/without VEGF-A (human recombinant VEGF-A [165]; 40 ng/ml; Peprotech) and heparin (2 U/ml), and injected s.c. at the abdomen of *Kif13b^iECKO* and

*Kif13b^WT*. 7 d after injection, Matrigel was isolated and analyzed by immunohistochemistry with endothelial marker CD31.

## Aortic ring assay

Aortic rings were isolated from tamoxifen-treated *Kif13b^iECKO* and *Kif13b^WT* as described (Baker et al, 2011). Aortic rings were embedded in the collagen matrix and incubated in Opti-MEM supplemented with 2.5% FBS and 30 ng/ml VEGF-A for 7 d. The rings and sprouts were fixed with 4% PFA.

## Syngraft tumor implantation

To monitor the tumor growth, B16F10-luc2 was injected at the right dorsal flank by s.c. The growth of s.c. injected tumor was measured

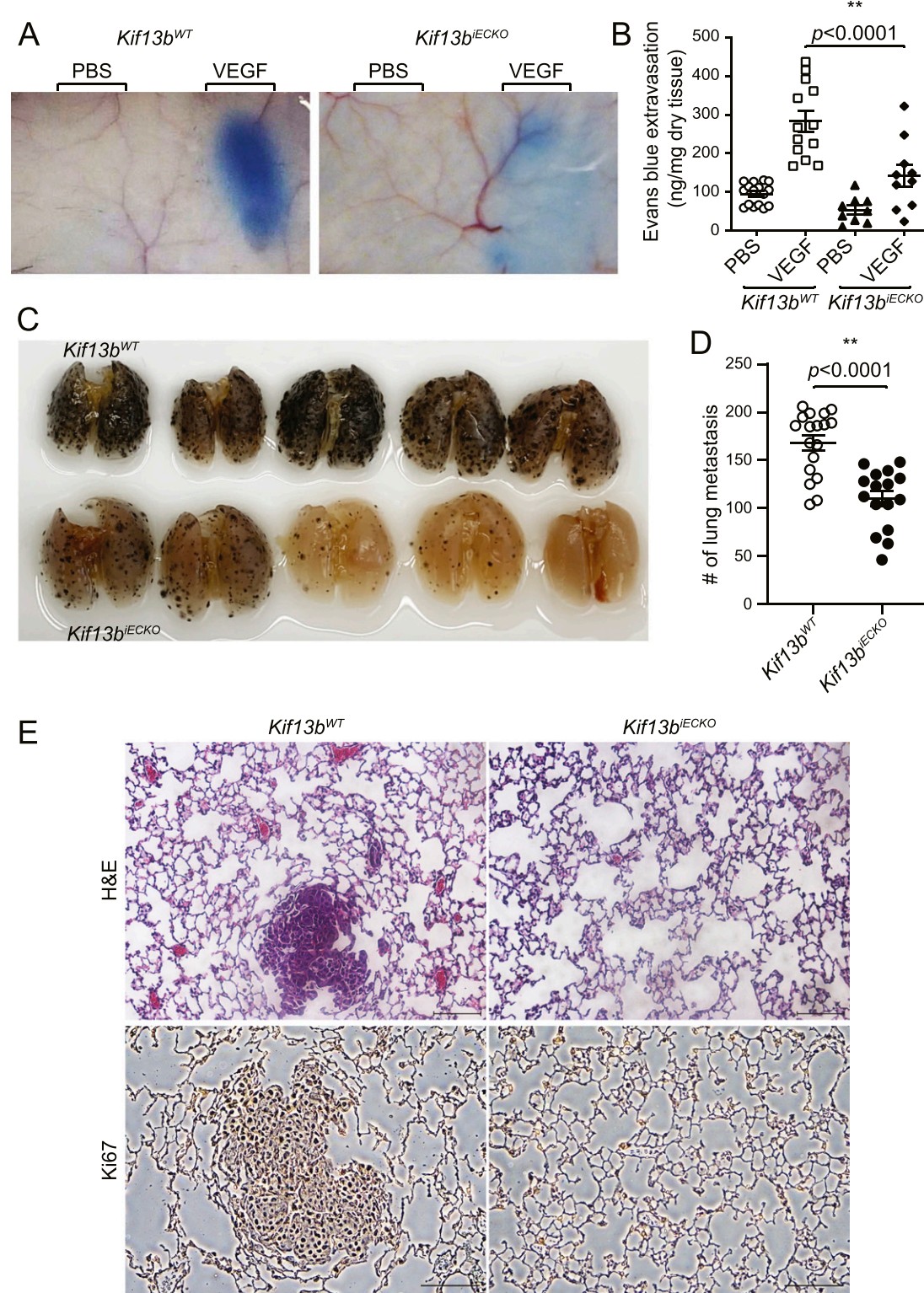

**Figure 4. Endothelial specific deletion of KIF13B prevents vascular leakage and metastasis in mice.**
**(A, B)** Evans blue extravasation showing VEGF-A–induced endothelial cell (EC) permeability in EC-specific *Kif13b* KO mice. *Kif13b^{iECKO}* and *Kif13b^{WT}* received i.v. injection of Evans blue followed by s.c. injection of VEGF-A or PBS to test VEGF-A–induced permeability. Extravasation of Evans blue to skin was measured and plotted in a graph. N = 13 and 9 for *Kif13b^{WT}* and *Kif13b^{iECKO}*, respectively. **(C, D)** Lung metastasis was examined by pigmented foci of B16F10 melanoma. *Kif13b^{WT}* and *Kif13b^{iECKO}* received tail vein injection with $1 \times 10^5$ of metastatic B16F10 melanoma cells. 2 wk after injection, the lungs were isolated and fixed. The pigmented foci on the lung surface were examined. The plots show the number of metastatic foci in the lung on day 13. *Kif13b^{iECKO}* had a markedly decreased rate of lung metastasis compared to the WT control. N = 18, 16 for *Kif13b^{WT}* and *Kif13b^{iECKO}*, respectively. **(E)** Representative image of H&E-stained lungs from *Kif13b^{WT}* and *Kif13b^{iECKO}* injected with $1 \times 10^5$ B16F10 melanoma via tail vein. Scale bar, 100 $\mu$m.

A

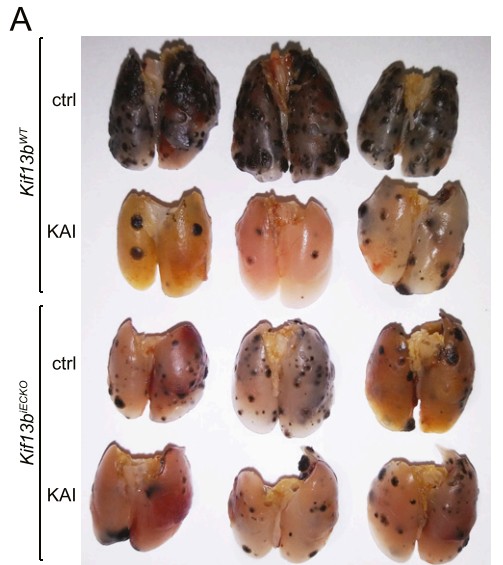

B

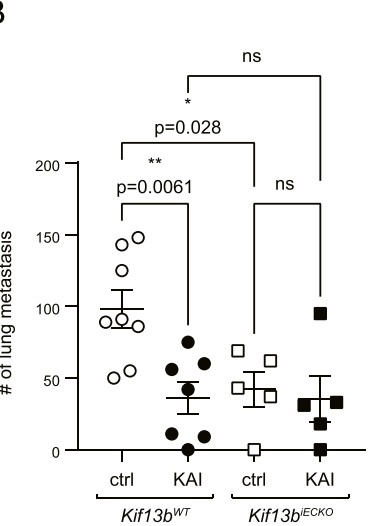

Figure 5.   **The peptide KAI targets KIF13B function in endothelial cells.**
**(A, B)** Cell-type specificity of KAI was examined by lung metastasis of B16F10 melanoma in $Kif13b^{WT}$ and $Kif13b^{iECKO}$. $Kif13b^{WT}$ and $Kif13b^{iECKO}$ received tail vein injection with $1 \times 10^5$ of metastatic B16F10 melanoma cells. Each genotype of mice was divided into two groups; each group received injections of control peptide or KAI (10 mg/kg b.w.) three times/week for 2 wk, and the lungs were isolated and fixed. The pigmented foci on the lung surface were examined. The plots show the number of metastatic lung foci on day 19. Treatment of KAI significantly decreased lung metastasis compared with the control peptide. N = 8, 7, 5, and 5 for $Kif13b^{WT}$+control peptide, $Kif13b^{WT}$ + KAI, $Kif13b^{iECKO}$+control peptide, and $Kif13b^{iECKO}$ + KAI, respectively.

by bioluminescence from B16F10-luc2 after injection of luciferin i.p. using the IVIS system (Perkin Elmer). The size of the s.c. tumor was measured by caliper and calculated as width × length × height/2.

To test the lung metastasis, $1 \times 10^5$ cells of B16F10 were injected into C57BL6 mice, $Kif13b^{WT}$, or $Kif13b^{iECKO}$ i.v. via tail vein. To test the efficacy of KAI, mice were divided into two groups, the control group received i.p injection of control peptide (10 mg/kg b.w.) and KAI group received KAI peptide (10 mg/kg b.w.) three times/week for 2 wk. 2 wk after injection, the lungs from mice were isolated, fixed, visually analyzed for the metastatic foci, and proceeded for the pathological analysis in the paraffin section.

### Immunohistochemistry

Tissues were fixed by normal buffer formalin, processed by a tissue processor, and embedded in the paraffin. Paraffin sections were deparaffinized, boiled in antigen retrieval buffer, and stained with antibodies.

### Evans blue endothelial albumin permeability assay

Evans blue 0.5% (Sigma-Aldrich) in saline was injected i.v. via the tail vein. 30 min after Evans blue injection, 100 ng VEGF-A or vehicle was injected intradermally on the back. Evans blue dye leakage in dorsal skin was assessed after 30 min, photographed, and extracted from the skin in formamide and quantified by spectrophotometry.

### Statistical analysis

Data were analyzed with GraphPad Prism 9 (GraphPad). Two samples were compared by Welch's *t* test, and two-tailed *P*-value was calculated. More than two samples were analyzed by one-way ANOVA, followed by Bonferroni's multiple comparisons test.

## Data Availability

This manuscript does not have large-scale data sets to deposit to the public databases.

## Supplementary Information

## Acknowledgements

Portions of this work were carried out in the Fluorescent Imaging Core and the Cardiovascular Research Core via the Research Resources Center (RRC) at the University of Illinois at Chicago (UIC). This work was supported by National Institutes of Health (NIH) 1R01EY029339-01A1 (KH Yamada), 1R56HL128342-01A1 (KH Yamada), Research to Prevent Blindness (RPB) Stein Innovation Award (AB Malik), and UIC Chancellor's Innovation Fund (KH Yamada).

### Author Contributions

SB Waters: data curation, formal analysis, and investigation.
JR Dominguez: data curation, formal analysis, and investigation.
H-D Cho: data curation, formal analysis, and investigation.
NA Sarich: data curation.
AB Malik: supervision, funding acquisition, and writing—review and editing.
KH Yamada: conceptualization, data curation, formal analysis, supervision, funding acquisition, validation, investigation, visualization, methodology, project administration, and writing—original draft, review, and editing.

## Conflict of Interest Statement

Authors have no conflict of interests that might be perceived to influence the results and/or discussion reported in this article.

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
