## [Reviewer comments · Life Science Alliance]

Life Science Alliance

KIF13B-Mediated VEGFR2 Trafficking is Essential for Vascular Leakage and Metastasis in vivo

Stephen Waters, Joseph Dominguez, Hyun-Dong Cho, Nicolene Sarich, Asrar Malik, and Kaori Yamada

DOI: <https://doi.org/10.26508/lsa.202101170>

Corresponding author(s): Kaori Yamada, University of Illinois at Chicago

Review Timeline:

Submission Date:	2021-07-23
Editorial Decision:	2021-08-13
Revision Received:	2021-09-16
Editorial Decision:	2021-10-12
Revision Received:	2021-10-13
Accepted:	2021-10-14

Transaction Report:

August 13, 2021

Re: Life Science Alliance manuscript #LSA-2021-01170-T

Dr. Kaori H. Yamada
University of Illinois at Chicago
Pharmacology
835 S. Wolcott, E403 MSB, MC 868
Chicago, IL 60612

Dear Dr. Yamada,

Thank you for submitting your manuscript entitled "KIF13B Mediated VEGFR2 Trafficking Regulates Vascular Leakage and Metastasis in vivo" to Life Science Alliance. The manuscript was assessed by expert reviewers, whose comments are appended to this letter. We invite you to submit a revised manuscript addressing the Reviewer comments.

Thank you for this interesting contribution to Life Science Alliance. We are looking forward to receiving your revised manuscript.

Sincerely,

B. MANUSCRIPT ORGANIZATION AND FORMATTING:

Reviewer #1 (Comments to the Authors (Required)):

In this manuscript, Waters et al have investigated the role endothelial KIF13B on VEGF-induced vascular leakage, VEGF-induced EC sprouting and tumor metastasis with Kif13b endothelial specific KO mice. Moreover, they have examined the previously established KAI peptide, which suppresses the binding KIF13B with VEGFR2. The experiments are carefully designed and well performed. Given the clinical potential of the KAI peptide, their results are interesting and important. However, the manuscripts contain various points to be improved before considering to publish by a journal. Specific comments are shown below.

1) I feel that each section of the introduction is not well connected logically. In addition, although the description of each data is proper, the connection between the data is weak. Furthermore, the description in the "introduction" and "discussion" sections are often redundant. On the other hand, necessary information in the discussion is missing. For instance, the authors described in the introduction "Thus, it can be a safer alternative strategy to target VEGF/VEGFR2 pathway for angiogenesis-related diseases such as cancer. " Safer than what? I guess the authors want to claim the potential advantage of KAI peptide for patient treatment than current clinical interventions. If so, the authors should mention VEGF inhibitors which is widely used in the clinics. The authors especially mentioned the importance of VEGFR2 trafficking on AKT and Src signaling. Does the treatment of ECs with KAI suppress those signaling pathways?

2) The authors showed the role of KIF13B on VEGF-induced endothelial sprouting, VEGF-induced vascular leakage, and tumor metastasis, which highlight different endothelial function. Design and result interpretation are fine, however, they do not connect nicely in the single manuscript from the logical point of view. How about discuss the connection of each data in the discussion section?

3) Abbreviations should be defined in the manuscript. For instance, KAI, IHC, and so on.

Additionally, the gene name should be described according to the mouse gene nomenclature. For instance, "KIF13B" should be "Kif13b"

4) The authors previously showed that KIF13B binds to VEGFR2 and controls VEGFR2 cell surface presentation. Then, they developed a peptide inhibitor which blocks binding of KIF13B with VEGFR2. What is the evidence that treatment of ECs with KAI suppresses VEGFR2 cell surface presentation?

5) In the method section, the authors must describe each information more precisely. What is the catalog number of each antibody? The name of animal ethics committees should be described. What is the dilution of Matrigel? What is the company provided VEGF? Mouse VEGF? Human VEGF? VEGF-A? VEGF-C? VEGF165? In the method section, VEGF concentration for matrigel plug assay was 40ng/ml. On the other hand, it is described as 50 ng/ml in the figure legend. What kind of statistical analysis the authors applied for each experiment?

6) Given the phenotype of EC specific Kif13b KO mice and KAI treated mice in the tumor metastasis assay, specificity of KAI peptide should be clarified. If KAI peptide is treated with EC specific Kif13b KO mice, what would be happened?

Reviewer #2 (Comments to the Authors (Required)):

Here, Waters et al. study the role for the kinesin-like motor protein KIF13B, which they previously have shown regulates intracellular transport of VEGFR2 to the cell surface. Using a newly generated endothelial-specific KIF13B deletion mouse model, the authors now show that in the absence of endothelial KIF13b, sprouting angiogenesis, lung colonization of B16F10 tumor cells injected in the tail vein, and vascular leakage are all suppressed. There is also delayed growth and vascularization of subcutaneous B16F10 tumors. The results phenocopy those seen with a KIF13B-derived peptide described by Dr. Yamada in several publications, which blocks the interaction between KIF13B and VEGFR2. This is a well conducted and presented study confirming an important role for VEGFR2 in control of the vascular barrier. It moreover highlights the potential of the KIF13B-based peptide in development of novel therapeutics to regulate pathological angiogenesis. I have no criticisms.

Reviewer #1:

1) I feel that each section of the introduction is not well connected logically. In addition, although the description of each data is proper, the connection between the data is weak. Furthermore, the description in the "introduction" and "discussion" sections are often redundant. On the other hand, necessary information in the discussion is missing. For instance, the authors described in the introduction "Thus, it can be a safer alternative strategy to target VEGF/VEGFR2 pathway for angiogenesis - related diseases such as cancer. "Safer than what? I guess the authors want to claim the potential advantage of KAI peptide for patient treatment than current clinical interventions. If so, the authors should mention VEGF inhibitors which is widely used in the clinics. The authors especially mentioned the importance of VEGFR2 trafficking on AKT and Src signaling. Does the treatment of ECs with KAI suppress those signaling pathways?"

Thank you for your helpful comments. We edited the text mainly between each paragraph to connect each paragraph logically. For example, between the 1st paragraph and 2nd paragraph of the introduction, we added, "Besides the total amount of VEGFR2 described above, the availability of VEGFR2 on the cell surface is a critical factor to regulate VEGFR2 signaling, as VEGFR2 receives VEGF on the cell surface. On the endothelial cell surface, ..."

As we thoroughly edited the text, we left track change in the manuscript.

Also, to describe what KAI is safer than, we wrote, "Concerning the systemic effect of anti-VEGF therapy (Ferrara and Adamis, 2016; Crawford and Ferrara, 2009), selectively targeting VEGFR2 trafficking can be a safer alternative strategy to target VEGF/VEGFR2 pathway for angiogenesis-related diseases such as cancer."

Regarding the signaling molecules under VEGF/VEGFR2, we revised them as "activation of downstream signaling such as SRC and ERK (Lanahan et al., 2010; Simons et al., 2016). "We have tested the effect of KAI on downstream signaling molecules such as SRC. However, we would like to keep the data for the next manuscript, as it is one of the most important data for the next one.

2) The authors showed the role of KIF13B on VEGF -induced endothelial sprouting, VEGF - induced vascular leakage, and tumor metastasis, which highlight different endothelial function. Design and result interpretation are fine, however, they do not connect nicely in the single manuscript from the logical point of view. How about discuss the connection of each data in the discussion section?

In the discussion section, we discussed the connection between endothelial permeability and angiogenesis. Please see the 4th paragraph and 5th paragraph in the discussion section.

3) Abbreviations should be defined in the manuscript. For instance, KAI, IHC, and so on. Additionally, the gene name should be described according to the mouse gene nomenclature. For instance, "KIF13B" should be "Kif13b"

In the revised manuscript, we carefully spell out the abbreviations when they first appear. Also, we followed the rule of the mouse gene nomenclature.

4) The authors previously showed that KIF13B binds to VEGFR2 and controls VEGFR2 cell surface presentation. Then, they developed a peptide inhibitor which blocks binding of KIF13B with VEGFR2. What is the evidence that treatment of ECs with KAI suppresses VEGFR2 cell surface presentation?

We showed the data in our previous publication (Yamada et al. 2017). As the description in the discussion was not clear, we revised it as "Our strategy is reducing the amount of VEGFR2 on the cell surface by inhibiting VEGFR2 trafficking mediated by KIF13B. Knockdown of KIF13B in EC did not affect the total amount of VEGFR2, whereas trafficking of VEGFR2 to the plasma membrane was inhibited (Yamada et al., 2014). Similarly, restoration of cell-surface VEGFR2 after its internalization was reduced by KAI treatment in cultured EC (Yamada et al., 2017). In the present study, EC-specific deletion of KIF13B did not affect the total amount of VEGFR2 (data not shown). Based on the in vitro study, inhibition of KIF13B by genetic depletion or KAI treatment only prevents trafficking of VEGFR2 to the plasmalemma and thereby blocked angiogenesis, suggesting that the primary role of KIF13B is in mediating the trafficking and spatial organization of VEGFR2 at the plasma membrane."

5) In the method section, the authors must describe each information more precisely. What is the catalog number of each antibody? The name of animal ethics committees should be described. What is the dilution of Matrigel? What is the company provided VEGF? Mouse VEGF? Human VEGF? VEGF - A? VEGF - C? VEGF165? In the method section, VEGF concentration for matrigel plug assay was 40ng/ml. On the other hand, it is described as 50 ng/ml in the figure legend. What kind of statistical analysis the authors applied for each experiment?

We added the catalog number of the antibodies in the methods section. Note, for the data in this manuscript, we used the rabbit polyclonal antibody against KIF13B from Sigma (HPA025-023), which was discontinued. The

new antibody from Sigma with a different lot number does not work well. We are currently using homemade mouse monoclonal antibody for recent data.

We added the name of the animal ethics committee in the methods section, "*Animal Care Committee administered through the Office of Animal Care and Institutional Biosafety at UIC.*"

Human recombinant VEGF-A165 is from Peprotech, we used 40 ng/ml VEGF in Matrigel plug assay. We corrected the figure legend. Matrigel was used without dilution.

We used Graph Pad Prism 9 (GraphPad Software) for statistical analysis. We analyzed data by one-way ANOVA followed by post hoc Bonferroni multiple comparisons. To compare two samples, we used Student's *t*-test. We added one paragraph of Statistical analysis in the methods section.

6) *Given the phenotype of EC specific Kif13b KO mice and KAI treated mice in the tumor metastasis assay, specificity of KAI peptide should be clarified. If KAI peptide is treated with EC specific Kif13b KO mice, what would be happened?*

Thank you for the great advice. We tested the effect of KAI treatment in lung metastasis in *Kif13b^{iECKO}* and *Kif13b^{WT}*. We added the data in Fig. 5. "*Consistent with Fig. 1, KAI treatment significantly reduced the number of metastasis in Kif13b^{WT} compared with treatment with ctrl peptide (Fig. 5A, B). Consistent with Fig. 4, Kif13b^{iECKO} and Kif13b^{WT} with ctrl peptide treatment showed a significant difference. However, KAI treatment in Kif13b^{iECKO} did not show further inhibition, compared with ctrl treatment in Kif13b^{iECKO} or KAI treatment in Kif13b^{WT} (Fig. 5A, B). This data suggests that the target of KAI is mainly KIF13B and VEGFR2 in EC.*" Then, we discussed in the discussion section, "*Metastasis is multiple-step events, such as escape of cancer cells from the primary tumor, intravasation of cancer cells into the bloodstream, survival of cancer cells in the bloodstream, extravasation of cancer cells through vascular barrier at the distal site, and colonization of cancer cells at the distal site. This experimental setting skips initial steps and starts from the injection of the cancer cells into the bloodstream. And the inhibitory effect of KAI was observed only in the presence of KIF13B in EC (Fig. 5), suggesting the main target of KAI is KIF13B-mediated VEGFR2 trafficking in EC, not survival of cancer cells in the bloodstream nor colonizing of cancer cells in the lungs.*"

Reviewer #2:

Here, Waters et al. study the role for the kinesin - like motor protein KIF13B, which they previously have shown regulates intracellular transport of VEGFR2 to the cell surface. Using a newly generated endothelial - specific KIF13B deletion mouse model, the authors now show that in the absence of endothelial KIF13b, sprouting angiogenesis, lung colonization of B16F10 tumor cells injected in the tail vein, and vascular leakage are all suppressed. There is also delayed growth and vascularization of subcutaneous B16F10 tumors. The results phenocopy those seen with a KIF13B - derived peptide described by Dr. Yamada in several publications, which blocks the interaction between KIF13B and VEGFR2. This is a well conducted and presented study confirming an important role for VEGFR2 in control of the vascular barrier. It moreover highlights the potential of the KIF13B - based peptide in development of novel therapeutics to regulate pathological angiogenesis. I have no criticisms.

Thank you for very positive comments.

October 12, 2021

RE: Life Science Alliance Manuscript #LSA-2021-01170-TR

Dr. Kaori H. Yamada
University of Illinois at Chicago
Pharmacology
835 S. Wolcott, E403 MSB, MC 868
Chicago, IL 60612

Dear Dr. Yamada,

Thank you for submitting your revised manuscript entitled "KIF13B Mediated VEGFR2 Trafficking Regulates Vascular Leakage and Metastasis in vivo". We would be happy to publish your paper in Life Science Alliance pending final revisions necessary to meet our formatting guidelines, as well as addressing the remaining points made by Reviewer #1.

- please use the [10 author names, et al.] format in your references (i.e. limit the author names to the first 10)
- the author contribution assigned to Asrar B Malik is limited. If any of these Contributor roles apply, please add them: <https://casrai.org/CRedit/>

A. FINAL FILES:

B. MANUSCRIPT ORGANIZATION AND FORMATTING:

****It is Life Science Alliance policy that if requested, original data images must be made available to the editors. Failure to provide**

original images upon request will result in unavoidable delays in publication. Please ensure that you have access to all original data images prior to final submission.**

The license to publish form must be signed before your manuscript can be sent to production. A link to the electronic license to publish form will be sent to the corresponding author only. Please take a moment to check your funder requirements.

Sincerely,

Reviewer #1 (Comments to the Authors (Required)):

According to previous my comments, the authors address the raised issues. Importantly, data in Figure 5 strengthen the authors' conclusion. Date is fine. The discussion part was improved very much. However, I still feel the introduction section is not connected well and should be improved. I suggest to ask a specialist in the field to edit the introduction part. The information regarding statistical analysis is missing. What kind of statistical analysis was applied for each experiment? This is the serious problem. Those points must be fixed before publication.

Minor points

- 1) What is "VEGF-related diseases"?
- 2) Endothelial cell? Or EC?
- 3) The authors defined vascular endothelial growth factor A as "VEGF-A". However, in the most of the part of the manuscript, the authors' described as VEGF.

October 14, 2021

RE: Life Science Alliance Manuscript #LSA-2021-01170-TRR

Dr. Kaori H. Yamada
University of Illinois at Chicago
Pharmacology
835 S. Wolcott, E403 MSB, MC 868
Chicago, IL 60612

Dear Dr. Yamada,

Thank you for submitting your Research Article entitled "KIF13B-Mediated VEGFR2 Trafficking is Essential for Vascular Leakage and Metastasis in vivo". It is a pleasure to let you know that your manuscript is now accepted for publication in Life Science Alliance. Congratulations on this interesting work.

DISTRIBUTION OF MATERIALS:

Again, congratulations on a very nice paper. I hope you found the review process to be constructive and are pleased with how the manuscript was handled editorially. We look forward to future exciting submissions from your lab.

Sincerely,
